# ViR: Vision Retention Networks

## Abstract

Vision Transformers (ViTs) have attracted a lot of popularity in recent years, due to their exceptional capabilities in modeling long-range spatial dependencies and scalability for large scale training. Although the training parallelism of self-attention mechanism plays an important role in retaining great performance, its quadratic complexity baffles the application of ViTs in many scenarios which demand fast inference. This effect is even more pronounced in applications in which autoregressive modeling of input features is required. In Natural Language Processing (NLP), a new stream of efforts have proposed parallelizable models with recurrent formulation that allows for efficient inference in generative applications. Inspired by this trend, we propose a new class of computer vision models, dubbed Vision Retention Networks (ViR), with dual parallel and recurrent formulations, which strike an optimal balance between fast inference and parallel training with competitive performance. In particular, ViR scales favorably for image throughput and memory consumption in tasks that require higher-resolution images due to its flexible formulation in processing large sequence lengths. The ViR is the first attempt to realize dual parallel and recurrent equivalency in a general vision backbone for recognition tasks. We have validated the effectiveness of ViR through extensive experiments with different dataset sizes and various image resolutions and achieved competitive performance. Our code and pretrained models will be made publicly available.

## 1 Introduction

During the recent years, Transformers (Vaswani et al., 2017) and their variants (Devlin et al., 2019; Dosovitskiy et al., 2020) have shown competitive performance across multiple domains such as Natural Language Processing (NLP) and Computer vision. The main building block of Transformers is self-attention which allows for cross interaction among all input sequence tokens with each other. This scheme is effective in capturing both short and long-range spatial dependencies but also imposes time and space quadratic complexity in terms of the input sequence length. The training parallelism of Transformers allow for competitive performance. However, the inference is slow and expensive due to the computational complexity.

Recently, Retentive Network (RetNet) (Sun et al., 2023) and Receptance Weighted Key Value (RWKV) (Peng et al., 2023) independently proposed novel model architectures that include the training parallelism of transformers and fast recurrent inference. The RWKV model uses a linear channel-wise attention to relax the pairwise dot product bottleneck of vanilla self-attention. The RetNet on the other hand proposes the concept of retention with dual form parallel and recurrent representations. It is noteworthy to mention that both RWKV and RetNet models are primarily proposed for autoregressive text generation.

Although Convolutional Neural Networks (CNNs) have been commonly used as the de-facto architecture for various applications, the introduction of Vision Transformers (Dosovitskiy et al., 2020) (ViT) demonstrated the possibility of achieving State-of-the-Art (SOTA) performance with a similar model to the Transformers for NLP applications. As opposed to the autoregressive formulation in which tokens from left to right are processed at each step to predict the next value, ViT uses the entire token representations.

In the case of long token sequences (*e.g.* high-resolution images), processing the entire tokens may create a bottleneck due to the quadratic complexity of the self-attention layers. As a result, despite the

competitive performance of ViT models, this limits their usage for applications that require real-time processing of high-resolution images (*e.g.* autonomous vehicles).

In this work, inspired by the success of RetNet, we explore the possibility of leveraging the duality of parallel and recurrent formulations to enable fast and memory efficient deployment while maintaining the training parallelism with a competitive performance.

In particular, the combination of parallel and recurrent modes, referred to as chunk-wise formulation, enables optimal combination of both modes based on specific run-time hyper-parameters (*e.g.* batch size) and hardware requirements. Due to this formulation, the memory consumption in ViR model can then be decoupled from the sequence length, hence making it easier to process high-resolution images in an efficient manner.

In order to improve the efficiency, we have redesigned the retention mechanism by removing the gated function. In addition, the proposed retention formulation is also generic and does not rely on any specific relative position embedding formulations (*e.g.* xPos (Sun et al., 2022) as in RetNet. Our proposed ViR is the first attempt beyond generative applications for leveraging autoregressive vision-friendly retentive networks for recognition tasks (*e.g.* image classification)

The summary of our specific contributions in this work is as follows:

- We introduce ViR, which is the first attempt in leveraging autoregressive retentive network with dual parallel and recurrent formulations for vision recognition tasks. We demonstrate that ViR can scale favorably to larger image resolutions in terms of image throughput and memory consumption.
- We propose a general vision backbone with a redesigned retention mechanism. The new retention mechanism is free of any gating function and does not rely on any specific relative position embedding formulations.
- We have validated the effectiveness of ViR by pretraining and finetuning on both ImageNet-21K and ImageNet-1K datasets for different models sizes to demonstrate the scalability of our proposed model as a general computer vision backbone.

## 2 RELATED WORK

**Vision Transformers** ViT (Dosovitskiy et al., 2020) introduced a new paradigm to move away from the convolutional inductive biases towards a simpler model with minimal priors. The effectiveness of self-attention in modeling long-range spatial dependencies and scalability of ViTs make them a great candidate as a backbone model for various vision tasks. However, the quadratic complexity of self-attention creates a bottleneck for fast deployment, especially for high-resolution images with longer sequence lengths. Swin Transformers (Liu et al., 2021) proposed to compute self-attention in smaller partitioned windows to address this problem.

Although this scheme improves the efficiency, the limited cross-region interactions across local windows may impact the performance. Independently, Pyramid Vision Transformer (PVT) (Wang et al., 2021) introduced a hierarchical architecture, similar to Swin Transformer, that employ a patch embedding layer at the beginning of each stage and reduces the spatial dimension to improve the computational efficiency.

On the other hand, Twins Transformer (Chu et al., 2021a) introduced a spatially separable self-attention mechanism that consisted of global sub-sampling and locally-grouped modules that can model both short and long-range interactions in an efficient manner. Several follow up efforts proposed to address this issue by introducing global (Hatamizadeh et al., 2023b) or carrier (Hatamizadeh et al., 2023a) tokens and multi-axis grid attention (Tu et al., 2022).

In addition to these works, a stream of hybrid models (*i.e.* CNN and ViT) (Graham et al., 2021; Wu et al., 2021; Yuan et al., 2021) were proposed to improve the data efficiency and achieve competitive performance without considerably larger model sizes. Convolutional vision Transformer (CvT) (Wu et al., 2021) proposes the concept of convolutional token embedding layer which is integrated with a Transformer block in a hierarchical architecture to improve the data efficiency and performance of the ViT models. In addition, Tokens-To-Token Vision Transformer (T2T-ViT) (Yuan et al., 2021) introduced a tailored transformation layer for aggregating nearby tokens which can be ten used as image priors for leveraging spatial correlations.

Cross-covariance Image Transformer (XCiT) (Ali et al., 2021) proposed a transposed self-attention block for capturing the token interactions in feature channels space. In addition, by conditioning the position encoding on localized patch tokens, Conditional Position encoding Vision Transformer (CPVT) (Chu et al., 2021b) achieved better performance on different recognition tasks such as image classification and object detection. Our proposed contributions in this work are orthogonal to these recent advances as ViR can benefit from a hybrid architecture as well as a window-based retention. Please see Sec. 5.3 for discussion on the effect of hybrid architectures on the performance of ViR models.

**Autoregressive Models** Deep Autoregressive models Greff et al. (2016); Van Den Oord et al. (2016); Van den Oord et al. (2016); Chen et al. (2018); Radford et al. (2018) have primarily been used for generative application and achieved great success in this domain. Most notably, PixelCNN (Van den Oord et al., 2016) and PixelRNN (Van Den Oord et al., 2016) demonstrated that sequential pixel-by-pixel prediction can be an effective in learning the explicit probability distribution for both discrete and continuous data while having better training stability compared to Generative Adversarial Networks (GANs) (Goodfellow et al., 2014). With the emergence of Transformers (Vaswani et al., 2017), several efforts (Parmar et al., 2018; Chen et al., 2020; Cao et al., 2021; Chang et al., 2022) demonstrated the capability of autoregressive modeling at scale. However, the sequential nature of autoregressive decoding, which requires access to previously generated tokens for future predictions, hinders the efficiency of such models.

**Self-attention Alternatives** To address the quadratic computation complexity of self-attention, many efforts have proposed various approaches such as approximation of the $\mathrm{softmax}$ activation function (Joulin et al., 2017; Gao et al., 2020), linear attention by using other kernels (Wang et al., 2020; Katharopoulos et al., 2020a) to estimate the attention scores or computing the attention in the channel feature space (Ali et al., 2021). However, the improved efficiency negatively impacts the performance of the model. Other efforts (Zhai et al., 2021; Gu et al., 2021) have also proposed to entirely replace the self-attention with other mechanisms.

In particular, recently in NLP, RWKV (Peng et al., 2023) and RetNet (Sun et al., 2023) proposed to redefine the Transformers to leverage the duality of parallel and recurrent formulation for training and inference. RWKV follows an attention-free formulation (Zhai et al., 2021) but employs an exponential decay to enable the recurrent formulation. RetNet proposes to use multi-scale gated retention to maintain the expressivity of the contextual information and achieve competitive performance. Although our work is inspired by RetNet, it is aimed for computer vision, in particular recognition, and has a tailored retention mechanism and architecture redesign for optimal performance.

## 3 METHODOLOGY

### 3.1 RETENTION MECHANISM

In this section, we discuss the retention mechanism and its different formulations (Sun et al., 2023). Consider an input sequence $\mathbf{X} \in \mathbb{R}^{|X| \times D}$ that will be encoded in an autoregressive manner. Given the query ($\mathbf{q_n}$), key ($\mathbf{k_n}$) and value ($\mathbf{v_n}$) in state $\mathbf{s_n}$, this sequence-to-sequence mapping can be written as

$$
\begin{aligned}
\mathbf{s_n} &= \alpha \mathbf{s_{n-1}} + \mathbf{k_n}^\top \mathbf{v_n}, \\
\mathrm{Ret}(\mathbf{X_n}) &= \mathbf{q_n s_n},
\end{aligned}
\tag{1}
$$

where $\mathrm{Ret}$ and $\alpha$ denote retention and decay mask, respectively. In essence, $\mathbf{s_n}$ conveniently maintains the previous internal states. As shown in (Sun et al., 2023), retention can also be defined in a parallel formulation

$$
\mathrm{Ret}(\mathbf{X}) = (\mathbf{q}\mathbf{k}^\top \odot \mathbf{M})\mathbf{v},
\tag{2}
$$

Where $\mathrm{M}$ denotes a mask with a decay factor $\alpha$ as in

$$
\mathbf{M_{ij}} = \begin{cases} \alpha^{i-j}, & i \geqslant j \\ 0, & i < j \end{cases}
\tag{3}
$$

This dual representation of the retention in parallel and recurrent modes enable many desired properties such as training parallelism and fast inference. For longer sequences the recurrent mode

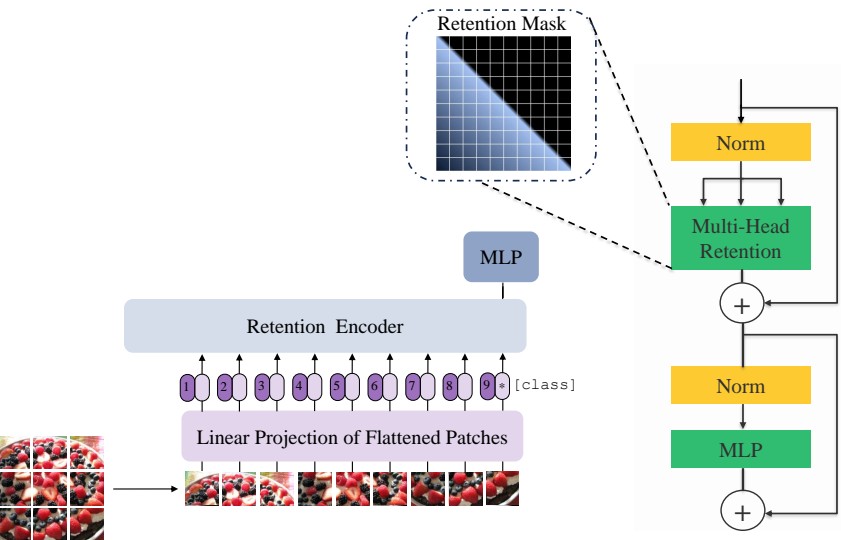

Figure 1: Overview of the architecture of ViR model. Similar to ViT, Flattened patches are linearly projected into a patch embedding. The position embedding are then added to the patch embedding and a class token is appended to this sequence. The retention encoder comprises of alternating Multi-Head Retention and MLP blocks. The MHR blocks use a causal decay mask. Best viewed in color.

can become inefficient. As a result, a hybrid approach, referred to as chunkwise, which combines recurrent and parallel formulation is desired. Specifically, the input $\mathbf{X}$ is split into smaller sequences with chunksize $C$, in which $\mathbf{x}_{[m]} = [\mathbf{x}_{(m-1)C+1}, \cdots, \mathbf{x}_{mC}]$ represents the $m$-th chunk. The chunkwise query, key and values can be defined as

$$\mathbf{q}_{[m]} = \mathbf{q}_{Cm:C(m+1)}, \quad \mathbf{k}_{[m]} = \mathbf{k}_{Cm:C(m+1)}, \quad \mathbf{v}_{[m]} = \mathbf{v}_{Cm:C(m+1)}, \tag{4}$$

The chunkwise retention formulation is as follows

$$\mathbf{R}_m = \mathbf{k}_{[m]}^\top (\mathbf{v}_{[m]} \odot \zeta) + \gamma^{\mathbf{B}} \mathbf{R}_{m-1}, \quad \zeta_{mt} = \gamma^{\mathbf{B}-m-1}$$

$$\mathrm{Ret}(\mathbf{X}_{[m]}) = (\mathbf{q}_{[m]} \mathbf{k}_{[m]}^\top \odot \mathbf{M})\mathbf{v}_{[m]} + (\mathbf{q}_{[m]} \mathbf{R}_{m-1}) \odot \xi, \quad \xi_{mt} = \alpha^{m+1} \tag{5}$$

The underlying motivation of the chunkwise formulation is to employ the parallel mode in each chunk, while processing cross-chunk representations in the recurrent mode. For high resolution images with long sequences, the chunkwise formulation allows for faster processing of tokens and decoupling the memory. In Sec. 5.2, we demonstrate how ViRs compare more favorably to ViTs due to the chunkwise formulation for efficient processing of longer sequences.

## 3.2 ViR MODEL

In the following, we discuss the components of ViR in more details. Fig. 1 illustrates an overview of our proposed model. Given an input image $\mathbf{X} \in \mathbb{R}^{H \times W \times C}$ with height $H$ and width $W$, it is partitioned into patches and flattened into a sequence of tokens. This is similar to the tokenization scheme which was previously proposed by ViT (Dosovitskiy et al., 2020). The tokenized patches are then projected into a patch embedding $Z = [\boldsymbol{z}_1, \cdots, \boldsymbol{z}_{|z|}] \in \mathbb{R}^{|z| \times D}$ with dimension $D$. Different from ViT, we first add the position embedding to the patch embedding and then append a [class] token ($\mathbf{Z}_n^0 = \mathbf{X}_{\mathrm{class}}$).

The output of the ViR encoder with $L$ layers ($\mathbf{Z}_L^n$) is used in a classification Mult-Layer Perceptron (MLP) head during both pre-training and finetuning. Due to the autoregressive nature of the ViR model, the position of the [class] plays an important role as appending to the end of embedding sequence acts as a summarizing of all the previous tokens.

In lieu of self-attention, we use retention to enforce a recurrent formulation via masking. However, our formulation does not depend on gated retention or specific relative position embeddings (*e.g.*

xPos (Sun et al., 2022) or RoPE (Su et al., 2021)) and achieves numerical equivalency between parallel, recurrent and hybrid (*i.e.* mixture of local recurrent and global parallel) formulations. Specifically, the parallel retention formulation solely depends on query $\mathbf{q}$, key $\mathbf{k}$, value $\mathbf{v}$ and a decay Mask $M$ and defined according to

$$\mathbf{q}, \mathbf{k}, \mathbf{v} = \mathbf{z}\mathbf{A}_{qkv} \tag{6}$$

$$\text{Ret}(\mathbf{z}) = (\frac{\mathbf{q}\mathbf{k}^{\top}}{\sqrt{D_h}} \odot \mathbf{M})\mathbf{v} \tag{7}$$

where $\text{Ret}$ represents retention and $D_h$ is a scaling factor to balance the compute and parameter counts. Note that the retention formulation is free of $\text{softmax}$ activation function which is commonly used in self-attention to improve performance and maintain training stability at the cost of reduced efficiency. In addition, the original retention formulation, as proposed in RetNet (Sun et al., 2023), increases the number of parameters due to the addition of the learnable gated function, and a result decreases the image throughput under the same network layout.

The retention ($\text{Ret}$) is further extended to Multi-Head Retention (MHR). The retention is computed across each head with a constant decay factor and normalized with LayerNorm (Ba et al., 2016) (LN) according to

$$\mathbf{Y} = \text{LN}([\text{Ret}_1(\mathbf{z}); \text{Ret}_2(\mathbf{z}); \cdots \text{Ret}_k(\mathbf{z})]) \tag{8}$$

A GELU activation function is then employed on the concatenated outputs and before projecting them with a linear layer

$$\text{MHR}(\mathbf{z}) = \text{GELU}(\mathbf{Y})\mathbf{A}_{mhr} \tag{9}$$

We use alternating MHR and MLP blocks with LayerNorm (LN) and residual connections as the building blocks of the encoder according to

$$\begin{aligned} \mathbf{Z}'^{l} &= \text{MHR}(\text{LN}(\mathbf{Z}^{l})) + \mathbf{Z}^{l-1} \\ \mathbf{Z}^{l} &= \text{MLP}(\text{LN}(\mathbf{Z}'^{l})) + \mathbf{Z}'^{l} \end{aligned} \tag{10}$$

## 4 EXPERIMENTS

### 4.1 SETUP

We trained all ViR model variants on ImageNet-1K dataset (Deng et al., 2009) except for ViR-L/14. This model was first pre-trained on ImageNet-21K dataset on $224 \times 224$ resolution. The pretraining was conducted for 90 epochs with a global batch size of 4096 and an initial learning rate of $1e^{-3}$ with a cosine decay learning rate scheduler. The model was subsequently finetuned on both $224 \times 224$ and $448 \times 448$ resolutions with a learning rate of $5e^{-5}$. In addition, the models on ImageNet-1K were trained for 600 epochs with a learning rate of $3e^{-3}$, weight decay of $5e^{-2}$ and global batch size of 4096.

We used moderate data augmentation techniques such as mixup and cutmix. For Hybrid ViR models, we used a 4-stage hierarchical architecture in which the first 2 stages comprise of residual CNN-based blocks, while the rest of stages contain ViR-based blocks. In between each stage, the resolution is decreased by a factor of two with strided CNN layers.

### 4.2 IMAGE CLASSIFICATION

We present image classification benchmarks for all models in Table 1. The ViR models demonstrate competitive performance across different model variants. Specifically, ViR variants outperform ViT counterparts by considerable margins different models, validating the effectiveness of our proposed appproach. The ViR-L/14 model also achieves competitive performance when pretrained and finetuned on ImageNet-21K and ImageNet-1K datasets, respectively.

Increasing the image resolution from $224 \times 224$ to $448 \times 448$ during the finetuning results in a considerable +1.1% improvement in terms of Top-1 accuracy. Hence, these benchmarks demonstrates the scalability of ViR models to larger training dataset and higher image resolutions.

Table 1: Image classification benchmarks on **ImageNet-1K** (Deng et al., 2009) validate set. Models with ‡ are pretrained on ImageNet-21K dataset.

| Method | Param (M) | FLOPs (G) | Image Size | Top-1 (%) |
|---|---|---|---|---|
| ResMLP-S12 (Touvron et al., 2021a) | 15 | 3.0 | $224^2$ | 76.6 |
| PVT-v2-B1 (Wang et al., 2022) | 13 | 2.1 | $224^2$ | 78.7 |
| GC ViT-XXT (Hatamizadeh et al., 2023b) | 12 | 2.1 | $224^2$ | 79.8 |
| DeiT-Small/16 (Touvron et al., 2021b) | 22 | 4.6 | $224^2$ | 79.9 |
| T2T-ViT-14 (Yuan et al., 2021) | 22 | 5.2 | $224^2$ | 81.5 |
| CPVT-Small-GAP (Chu et al., 2021b) | 23 | 4.6 | $224^2$ | 81.5 |
| ResNet50 (He et al., 2016) | 25 | 4.1 | $224^2$ | 76.1 |
| CrossViT-S (Chen et al., 2021) | 26 | 5.6 | $224^2$ | 81.0 |
| PVT-Small (Wang et al., 2021) | 24 | 3.8 | $224^2$ | 79.8 |
| Twins-PCPVT-S (Chu et al., 2021a) | 24 | 3.8 | $224^2$ | 81.2 |
| Swin-T (Liu et al., 2021) | 29 | 4.5 | $224^2$ | 81.3 |
| CoAtNet-0 (Dai et al., 2021) | 25 | 4.2 | $224^2$ | 81.6 |
| PVT-v2-B2 (Wang et al., 2022) | 25 | 4.0 | $224^2$ | 82.0 |
| ConvNeXt-T (Liu et al., 2022b) | 29 | 4.5 | $224^2$ | 82.1 |
| Focal-T (Yang et al., 2021) | 29 | 4.9 | $224^2$ | 82.2 |
| CSwin-T (Dong et al., 2022) | 23 | 4.3 | $224^2$ | 82.7 |
| ResNet-101 (He et al., 2016) | 44 | 7.9 | $224^2$ | 77.4 |
| ResMLP-S24 (Touvron et al., 2021a) | 30 | 6.0 | $224^2$ | 79.4 |
| PVT-Medium (Wang et al., 2021) | 44 | 6.7 | $224^2$ | 81.2 |
| T2T-ViT-19 (Yuan et al., 2021) | 39 | 8.9 | $224^2$ | 81.9 |
| Twins-PCPVT-B (Chu et al., 2021a) | 44 | 6.7 | $224^2$ | 82.7 |
| Swin-S (Liu et al., 2021) | 50 | 8.7 | $224^2$ | 83.0 |
| ConvNeXt-S (Liu et al., 2022b) | 50 | 8.7 | $224^2$ | 83.1 |
| PVT-v2-B3 (Wang et al., 2022) | 45 | 6.9 | $224^2$ | 83.2 |
| ViT-L/32 (Dosovitskiy et al., 2020) | 328 | 15.3 | $224^2$ | 71.2 |
| ViT-B/32 (Dosovitskiy et al., 2020) | 88 | 4.4 | $224^2$ | 73.4 |
| ViT-L/16 (Dosovitskiy et al., 2020) | 304 | 59.7 | $224^2$ | 76.5 |
| ResNet-152 (He et al., 2016) | 60 | 11.6 | $224^2$ | 78.3 |
| ViT-B/16 (Dosovitskiy et al., 2020) | 86 | 17.6 | $224^2$ | 77.9 |
| ResMLP-B24 (Touvron et al., 2021a) | 116 | 23.0 | $224^2$ | 81.0 |
| PVT-Large (Wang et al., 2021) | 61 | 9.8 | $224^2$ | 81.7 |
| DeiT-Base/16 (Touvron et al., 2021b) | 86 | 17.6 | $224^2$ | 81.8 |
| CrossViT-B (Chen et al., 2021) | 104 | 21.2 | $224^2$ | 82.2 |
| T2T-ViT-24 (Yuan et al., 2021) | 64 | 14.1 | $224^2$ | 82.3 |
| CPVT-B (Chu et al., 2021b) | 88 | 17.6 | $224^2$ | 82.3 |
| Twins-PCPVT-L (Chu et al., 2021a) | 61 | 9.8 | $224^2$ | 83.1 |
| Swin-B (Liu et al., 2021) | 88 | 15.4 | $224^2$ | 83.3 |
| PVT-v2-B4 (Wang et al., 2022) | 62 | 10.1 | $224^2$ | 83.6 |
| Twins-SVT-L (Chu et al., 2021a) | 99 | 15.1 | $224^2$ | 83.7 |
| ConvNeXt-B (Liu et al., 2022b) | 89 | 15.4 | $224^2$ | 83.8 |
| ViT-L/16‡ (Dosovitskiy et al., 2020) | 86 | 17.6 | $224^2$ | 85.1 |
| PVT-v2-B5 (Wang et al., 2022) | 82 | 11.8 | $224^2$ | 83.8 |
| **ViR-B/32** | 88 | 4.3 | $224^2$ | 75.7 |
| **ViR-S/16** | 22 | 4.2 | $224^2$ | 78.3 |
| **Hybrid ViR-S/16** | 31 | 3.3 | $224^2$ | 80.3 |
| **ViR-B/16** | 86 | 16.8 | $224^2$ | 81.3 |
| **Hybrid ViR-B/16** | 75.8 | 8.8 | $224^2$ | 82.4 |
| **ViR-L/14‡** | 304 | 77.8 | $224^2$ | 84.9 |
| **ViR-L/14‡** | 304 | 310.3 | $448^2$ | 86.0 |

# 5  ABLATION

## 5.1  COMPONENT STUDY

In this section, we study the effect of different component design choices on the overall performance by examining the Top-1 and throughput trade-off. As the base model, we use a ViR-B/16 with a Top-1 accuracy of 81.3% on ImageNet-1K dataset.

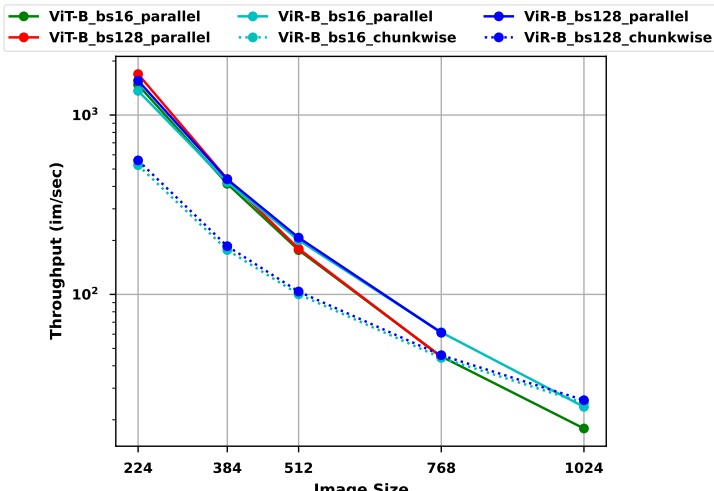

Figure 2: Comparison of image throughput for ViT-B and ViR-B networks. Throughput is measured on an A100 80GB NVIDIA GPU with batch sizes of 16 and 128. With batch size of 128, the parallel mode went OOM for both ViT and ViR. At the 768 image size, chunkwise matches the throughput of parallel mode, and is also the only configuration capable of processing 128 batch size at 1024 resolution.

First, we studied the effect of `[class]` token by removing it and using a global average pooling layer before the classification head. In this case, the Top-1 accuracy decreases by 0.4%. As discussed in Sec.3.2, the `[class]` plays an important role as it encapsulates global information from the preceding tokens that can be useful for the task of image classification. In addition, the throughput decreased by 1.90%.

We also investigated the effect of adding a gated function to the retention. For fair comparison, we reduced the number of layers to match the same number of parameters as the base model. However, this configuration decreased the image throughput and Top-1 accuracy by 2.91% and 0.3% respectively. Furthermore, we replaced the proposed GELU activation function with a Swish activation function, as originally proposed in RetNet. This configuration slightly decreased the image throughput by 1.04% while also lowering the Top-1 accuracy by 0.2%.

We also investigated effect of scaling the key tensor, in lie of the query. Although image throughput and Top-1 accuracy remain roughly unchanged, we observed some instabilities with sudden changes in loss values during training. In addition, as opposed to an autoregressive formulation, we also studied the possibilities of using multipass encoding by providing both left and right token orders.

Our results show that although Top-1 accuracy is slightly improved by +1.0%, the throughput is severely impacted and reduced by half. Hence, multipass encoding does not provide an optimal performance vs. efficiency tradeoff in our case.

| Design Component | Throughput (im/sec) | Top-1 (%) |
|---|---|---|
| No class token | 1525 | 80.9 |
| Gated retention | 1516 | 81.0 |
| Swish activation | 1538 | 81.1 |
| Key (**k**) scaling | 1550 | 81.2 |
| Multipass encoding | 774 | 81.4 |
| **Base Model** | 1554 | 81.3 |

Table 2: Ablation study on the effect of different design choices on ImageNet Top-1 accuracy vs throughput performance tradeoff. The throughput is measured on an A100 80GB NVIDIA GPU with a batch size of 128. The base model is ViR-B/16.

## 5.2 THROUGHPUT ANALYSIS

The primary motivation behind ViR is to find an attention formulation that allows for high inference throughput without sacrificing model quality. In (Sun et al., 2023) the authors provide a brief overview of attention methods, comparing scaling complexity, memory complexity, and resulting model quality,

to arrive at the conclusion that the RetNet formulation achieves the best results in the "impossible triangle" of (1) inference cost, (2) training parallelism, and (3) model quality.

Related to computer vision, the sequence length $N$ is derived from the input height $H$, width $W$, and patch size $P$ ((Dosovitskiy et al., 2020)), forming $N = \frac{HW}{P^2}$. Of note, because compute and memory complexity scales quadratically with sequence length, for regular attention, we see a scaling rule of $O\left(\frac{H^2 W^2}{P^4}\right)$, which strongly inhibits pursuing higher resolution image processing.

Typical methods for working around this involve eliminating global attention in favor of local attention ((Liu et al., 2022a), (Li et al., 2022)), approximating the attention matrix ((Choromanski et al., 2021), (Wang et al., 2020), (Kitaev et al., 2020)), or choosing a different formulation of attention that has better scaling behavior ((Katharopoulos et al., 2020b), (Bolya et al., 2022), (Sun et al., 2023)).

Adopting the RetNet formulation allows us to understand the inference cost in three different modes: Recurrent, Chunkwise, and Parallel. Because the recurrent formulation only depends on the previous token to compute the next, the compute complexity wrt the input is $O(N)$. Parallel mode can process all tokens simultaneously, but comes with the quadratic scaling complexity of $O(N^2)$.

Chunkwise is a hybrid mode where one chunk only depends on the previous chunk, and within a chunk, we adopt the parallel formulation. Let $C$ be the chunk size, then the number of chunks is $\lceil \frac{N}{C} \rceil$, the per-chunk complexity is $O(C^2)$, resulting in an overall complexity of $O\left(\frac{N}{C}C^2\right) = O(NC)$.

Since modern inference hardware is able to simultaneously perform numerous math operations, the chunkwise formulation is compelling because it allows us to trade-off saturating the compute hardware (larger $C$) with computational complexity (smaller $C$). In addition to throughput improvements, recurrent and chunkwise also adopt desirable memory properties. If the downstream application doesn't require patch features (e.g. a classification task), then the memory complexity for recurrent is $O(1)$ and for chunkwise is $O(C^2)$. If patch features are required, then it becomes $O(N)$ and $O(N + C^2)$ respectively.

It can be seen in figure 2 how throughput varies between ViT-B and ViR-B at different image sizes, and particularly how ViR shows better scaling characteristics as resolution increases. At very high resolution, only ViR-chunkwise is able to run on an A100 80GB NVIDIA GPU, as the parallel variants run out of memory.

Due to the different compute complexity scaling rules between parallel and chunkwise, it is apparent how chunkwise eventually matches parallel throughput, and then surpasses it at high resolution. Refer to appendix A.1 and figure S.1 for how scaling works for ViT-L and ViR-L variants. Unsurprisingly, parallel mode runs out of memory at lower resolution (768) whereas chunkwise is able to operate under all settings.

## 5.3 HYBRID ARCHITECTURE

Due to lack of inductive biases such as locality of weight sharing in CNNs, ViTs often require more training data or comprehensive data augmentation to achieve the same accuracy in relatively small to medium-sized datasets (*e.g.* ImageNet-1K). The proposed ViR also face the same challenges in such benchmarks. As a result, we have presented Hybrid ViR-S/16 and ViR-B/16 variants to demonstrate the feasibility of integrating CNN-based encoders with ViR.

As presented in Table 1, Hybrid ViR-S/16 (80.3%) outperforms the counterpart ViT-S/16 (78.3%) by a considerable +2.0% margin. Similarly, Hybrid ViR-B/16 (82.4%) surpasses the ViT-B/16 (81.3%) by +1.1% in terms of Top-1 accuracy. These results confirm the possibility of achieving highly competitive performance under in small-scale data regimes by combining CNN and ViR models. We leave investigation of more advanced hybrid architectures to future efforts.

## 6 OUTLOOK

In this work, we demonstrated the first attempt in leveraging autoregressive vision transformers, with dual parallel and recurrent representations, for image recognition tasks. We believe that the proposed ViR can be further explored for other applications such as dense prediction tasks in which ViTs struggle with high-resolution images due to the quadratic complexity of its self-attention layers.

Other tasks such as autoregressive image generation can also benefit from this new formulation that allows for fast inference of considerably longer token sequences.

## 7 CONCLUSION

In this work, we introduced a new class of computer vision models, referred to as Vision Retention Networks (ViR), with dual parallel and recurrent formulations. The equivalency of these formulations allow for desired properties such as training parallelism and fast inference while maintaining a great performance. In addition, a hybrid formulation, denoted as chunkwise, enables processing of longer sequences with considerably more efficient time and space complexities. We have trained and tested the proposed ViR on ImageNet-1K and ImageNet-21K datasets with different resolutions and achieved competitive performance. Hence, this validates the effectiveness of the proposed ViR in different data regimes and image resolutions. We believe the proposed ViR could be the foundation of a new class of efficient vision-friendly models that offer training and inference flexibility for a variety of applications.

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

# A APPENDIX

## A.1 THROUGHPUT ANALYSIS

Further results for the throughput analysis provided for ViT and ViR large models, along with a full table of results.

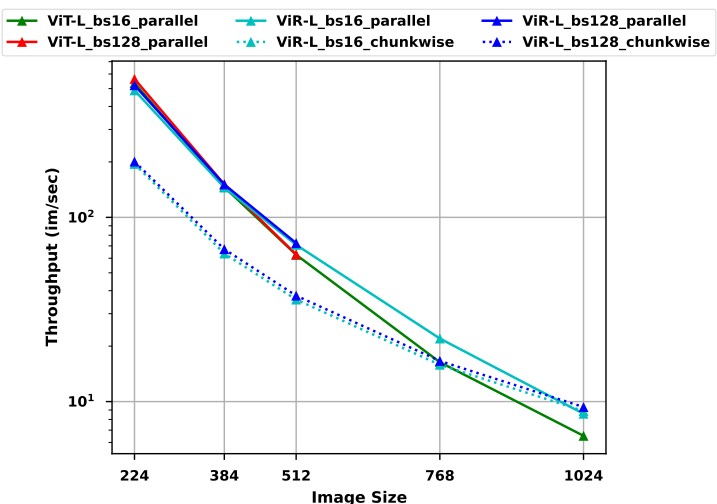

Figure S.1: Comparison of image throughput for ViT-L and ViR-L networks. Throughput is measured on A100 80GB GPU with batch sizes of 16 and 128. With $BS \in [768, 128]$, the parallel mode went OOM for both ViT and ViR. At the 768 image size, chunkwise matches the throughput of parallel mode, and is also the only configuration capable of BS=128 at 768 or 1024px.

Table S.1: Full throughput analysis results

| Model Type | Batch Size | Img Size | Mode | Throughput (im/sec) |
|---|---|---|---|---|
| vit-b | 16 | 224 | parallel | 1472.4778 |
| vit-b | 128 | 224 | parallel | 1691.0991 |
| vit-b | 16 | 384 | parallel | 413.6345 |
| vit-b | 128 | 384 | parallel | 439.6699 |
| vit-b | 16 | 512 | parallel | 176.5976 |
| vit-b | 128 | 512 | parallel | 179.8491 |
| vit-b | 16 | 768 | parallel | 45.0924 |
| vit-b | 128 | 768 | parallel | 44.8039 |
| vit-b | 16 | 1024 | parallel | 17.8864 |
| vit-b | 128 | 1024 | parallel | OOM |
| vir-b | 16 | 224 | parallel | 1363.4267 |
| vir-b | 16 | 224 | chunkwise | 525.3381 |
| vir-b | 128 | 224 | parallel | 1554.5852 |
| vir-b | 128 | 224 | chunkwise | 558.4554 |
| vir-b | 16 | 384 | parallel | 422.9626 |
| vir-b | 16 | 384 | chunkwise | 176.7866 |
| vir-b | 128 | 384 | parallel | 438.0182 |
| vir-b | 128 | 384 | chunkwise | 185.7153 |
| vir-b | 16 | 512 | parallel | 200.6345 |
| vir-b | 16 | 512 | chunkwise | 100.1816 |
| vir-b | 128 | 512 | parallel | 207.2452 |
| vir-b | 128 | 512 | chunkwise | 103.7436 |
| vir-b | 16 | 768 | parallel | 61.4616 |
| vir-b | 16 | 768 | chunkwise | 44.2865 |
| vir-b | 128 | 768 | parallel | 61.1587 |
| vir-b | 128 | 768 | chunkwise | 45.8153 |
| vir-b | 16 | 1024 | parallel | 23.6144 |
| vir-b | 16 | 1024 | chunkwise | 24.7994 |
| vir-b | 128 | 1024 | parallel | OOM |
| vir-b | 128 | 1024 | chunkwise | 25.7132 |
| vit-l | 16 | 224 | parallel | 536.0380 |
| vit-l | 128 | 224 | parallel | 564.4117 |
| vit-l | 16 | 384 | parallel | 146.0362 |
| vit-l | 128 | 384 | parallel | 151.0871 |
| vit-l | 16 | 512 | parallel | 62.6439 |
| vit-l | 128 | 512 | parallel | 62.8844 |
| vit-l | 16 | 768 | parallel | 16.3647 |
| vit-l | 128 | 768 | parallel | OOM |
| vit-l | 16 | 1024 | parallel | 6.5186 |
| vit-l | 128 | 1024 | parallel | OOM |
| vir-l | 16 | 224 | parallel | 489.3484 |
| vir-l | 16 | 224 | chunkwise | 195.1421 |
| vir-l | 128 | 224 | parallel | 521.3833 |
| vir-l | 128 | 224 | chunkwise | 200.8896 |
| vir-l | 16 | 384 | parallel | 145.2371 |
| vir-l | 16 | 384 | chunkwise | 63.6531 |
| vir-l | 128 | 384 | parallel | 150.8908 |
| vir-l | 128 | 384 | chunkwise | 67.2084 |
| vir-l | 16 | 512 | parallel | 70.9078 |
| vir-l | 16 | 512 | chunkwise | 35.8030 |
| vir-l | 128 | 512 | parallel | 72.4739 |
| vir-l | 128 | 512 | chunkwise | 37.5940 |
| vir-l | 16 | 768 | parallel | 21.9964 |
| vir-l | 16 | 768 | chunkwise | 15.8472 |
| vir-l | 128 | 768 | parallel | OOM |
| vir-l | 128 | 768 | chunkwise | 16.6237 |
| vir-l | 16 | 1024 | parallel | 8.5903 |
| vir-l | 16 | 1024 | chunkwise | 8.8889 |
| vir-l | 128 | 1024 15 | parallel | OOM |
| vir-l | 128 | 1024 | chunkwise | 9.3387 |