# OpenReview forum: "ViR: Vision Retention Networks"
_ICLR.cc/2024/Conference — ICLR 2024 Conference Withdrawn Submission_

### Official Review · Reviewer_pC11 · 2023-10-13

**Soundness:** 3 good
**Presentation:** 3 good
**Contribution:** 2 fair
**Rating:** 5
**Confidence:** 4

**Summary:**

The paper proposes vision retention networks, an alternative to vision transformers, that offer a dual parallel and recurrent formulation. The paper discusses in detail the mechanisms behind this new network and what are necessary changes to apply the retention architecture to vision. Through ablation studies they evaluate their architecture compared to possible alternatives. The authors successfully demonstrate how this architecture scales with more data and parameters, showing promising results. Finally, they demonstrate how the retention mechanism can lead to faster inference in cases, especially when encoding longer sequences, a consequence of evaluating larger images with the same patch size.

**Strengths:**

Proposing a new architecture involves numerous challenges.
1. The authors offer some ablations to demonstrate that the final architecture exhibits the best performance under some computational constraints. Experiments with models of different size and different datasets showcase how the architecture scales with the number of parameters and number of samples.
2. The authors exhibit how a chunkwise retention can lead to competitive inference in terms of throughput, while allowing inference with larger sequence lengths. Being able to trade-off saturating the compute hardware (larger C) with computational complexity (smaller C) seems like a great outcome of your work.
3. The code and models are going to be publicly available.

**Weaknesses:**

1. The main motivation is that retention networks can lead to faster inference. From experiments in Figure 2, Figure S1 and Table S1 this is not that evident. Keeping the patch size the same, the image size has to be increased substantially. For smaller -- typical -- images sizes, ViR models have smaller throughput compared to the regular ViT models.
2. The main advantage of retention networks is in the dual formulation. Since image encoding is not an autoregressive problem, it is not clear to me what are the benefits in this case (apart from chunkwise inference). Perhaps some relevant applications could include autoregressive image generation [1], or other decoder applications [2]. Still in most cases, having an encoder model for encoding image features makes the most sense.
3. Retention networks introduce "biases" due to the decay factor. It is not clear if applications that require more global features may suffer from these restrictions. The authors could perhaps try object detection or semantic segmentation as other suitable downstream tasks.
4. The retention mask introduces constraints due to the ordering of the tokens, i.e. patches. This is not discussed at all, apart from a short analysis on a multipass encoding as an alternative. Images are 2D, and it would make sense to decay the mask based on the 2D distance of the patches. This should be the main focus of the paper. Would be nice to also discuss connections with other "soft" inductive bias, e.g. see how ViTs can be adapted to consider distance between tokens at initialization in [3].

[1] Chen, Mark, et al. "Generative pretraining from pixels." International conference on machine learning. PMLR, 2020.

[2] Beyer, Lucas, et al. "A Study of Autoregressive Decoders for Multi-Tasking in Computer Vision." arXiv preprint arXiv:2303.17376 (2023).

[3] d’Ascoli, Stéphane, et al. "Convit: Improving vision transformers with soft convolutional inductive biases." International Conference on Machine Learning. PMLR, 2021.

**Questions:**

1. Are the baseline ViTs using flash attention [1] during inference? This can a big difference, especially for longer sequence lengths. In general, you could present only optimum (based on the best batch size) throughput for each image size, instead of showcasing results for multiple batch sizes. This would also improve the readability of Figure 2 (and Figure S1).
2. A lot of (very successful) work has been built on top of ViT to makes them faster for inference. A lot of them rely on merging tokens at different levels of the Transformer architecture, e.g. [2]. Is it still possible to apply something like this with your new architecture?
3. ViT numbers reported in Table 1 are suboptimal. Better ways to train ViT lead to higher accuracies [3, 4]. In general, models in Table 1 are trained for different number of samples and are difficult to compare with each other.
4. Text could use some improvement. For example, I don't really understand the Setup section. How does the Hybrid architecture exactly look like?
5. In [5] they also remove the [class] token but find significant computational benefits.
6. How does chunkwise compare to other methods such as model partitioning and activation checkpointing?

[1] Dao, Tri. "Flashattention-2: Faster attention with better parallelism and work partitioning." arXiv preprint arXiv:2307.08691 (2023).

[2] Bolya, Daniel, et al. "Token merging: Your vit but faster." arXiv preprint arXiv:2210.09461 (2022).

[3] Steiner, Andreas, et al. "How to train your vit? data, augmentation, and regularization in vision transformers." arXiv preprint arXiv:2106.10270 (2021).

[4] Touvron, Hugo, et al. "Training data-efficient image transformers & distillation through attention." International conference on machine learning. PMLR, 2021.

[5] Zhai, Xiaohua, et al. "Scaling vision transformers." Proceedings of the IEEE/CVF Conference on Computer Vision and Pattern Recognition. 2022.

---

### Official Review · Reviewer_98mC · 2023-10-30

**Soundness:** 2 fair
**Presentation:** 3 good
**Contribution:** 1 poor
**Rating:** 3
**Confidence:** 5

**Summary:**

The paper proposes an adaptation of the very recent retention network architecture to vision tasks, in particular for image classification. Retention Networks are a variant of transformer that is half way to RNN and pose some advantages to auto-regressive tasks. The proposed model adapts the architecture minimally (removing positional encoding / learnable gating function...) and trains models on ImageNet1k classification.

**Strengths:**

The paper focuses on a new and promising line of models.

The paper narrative is clear: the goal is stated clearly, and the experiments are well targeted.

The proposed models have wide applicability.

**Weaknesses:**

One of the main appeals of RetNets is that it can handle the auto-regressive training and use training parallelization (one single fwd pass with n tokens leads to n loss terms). The auto-regressive nature of the task is however lost in vision applications. Thus I do not see how this is a good fit, and how the arguments about training parallelism are valid.

Performance seems to be ok when compared to ViT, but slightly more advanced architectures like PVT already surpass the proposed architecture by wide margins. The proposed architecture is also based on a pyramidal design, so the more direct comparison is PVT rather than ViT.

Claims about throughput are not adequately tied to performance. We only see an isolated result in Table 1 with larger image sizes, but there are no comparisons to other methods with larger image size in terms of performance - only Fig. 2 compares throughput against ViT. However, ViT has been thoroughly improved in terms of throughput, and how it scales in terms of perf is not included in the paper.

I was expecting experiments beyond ImageNet1k - after all there isn't a lot of technical novelty so the empirical evaluation has to be top notch.

Minor:
"D_h is a scaling factor to balance the compute and parameter counts" D_h in eq 7 cannot have this effect. Is D_h used elsewhere?

**Questions:**

Performance against competition is a bit hard for the rebuttal - but maybe I misunderstood something that the authors would like to clarify. The throughput claims however have more margin to be expanded. And maybe the authors have some considerations on why this type of model is adequate for vision - and in particular why the "training parallelism" claim has an impact for a classification loss.

---

### Official Review · Reviewer_A5wE · 2023-10-31

**Soundness:** 3 good
**Presentation:** 3 good
**Contribution:** 2 fair
**Rating:** 3
**Confidence:** 4

**Summary:**

This work (ViR) adopts the newly emerging RetNet architecture on the image classification task. The authors tested the architecture on the image classification task with the ImageNet dataset and achieved better accuracy numbers than the vanilla visual transformers (ViT) of similar size and computes. The authors also conducted throughput studies between ViRs and ViTs and claims ViR can scale favorably to larger image resolutions in terms of throughput and memory consumption.

**Strengths:**

1. This work is the first few works if not the very first, applying the newly emerging RetNet architecture to the image classification task.
2. The accuracy numbers seem better than ViT with the controlled number of parameters and FLOPs.

**Weaknesses:**

1. The work partly focuses on inference performance, which is very worth studying since that’s one of the biggest benefits the retention structure could bring. However, the comparisons on the throughput are not detailed enough. The comparison is only conducted on a very large high-power GPU with a large memory size. It would be nice to show how the throughput changes w.r.t. Input size on small platforms such as GPUs with small memory. In addition to that, the throughput numbers are not only dependent on the algorithm design but also on many other moving parts such as the underlying implementations [C1].
2. The work is rather straightforward. The most advantages are inherited from the RetNet architecture. That is fine if enough reasonings and analysis are given for the specific task (here image classification) or studying the generality of multiple vision tasks are provided. At the moment they are lacking.
3. The authors claim in the contributions that this can be used as a general vision backbone. However, without testing on at least certain relative position critical tasks such as detection, it is too early to draw this conclusion. And it is not straightforward for RetNet.

[C1] FlashAttention: Fast and Memory-Efficient Exact Attention with IO-Awareness

**Questions:**

1. Could the authors provide an intuition despite the patches of an image are not treated equally, namely they are assigned with different decay values according to the ordering, ViR still outperforms ViT? In other words, if the patch with essential information is masked with a large decay, would the performance be worse? Artificially creating a test set might help to answer this question.
2. What is the chunk size used in Figure 2 for ViR chunkwise?

---

### Official Review · Reviewer_uppP · 2023-11-01

**Soundness:** 3 good
**Presentation:** 2 fair
**Contribution:** 2 fair
**Rating:** 3
**Confidence:** 2

**Summary:**

This paper is the pioneering work to apply the concept of retention with dual-form parallel and recurrent representations to Vision Transformers (ViTs). Specifically, ViTs suffer from the issue of extending to high-resolution images due to the quadratic complexity w.r.t the number of image tokens. So this work leverages the retention mechanism to enable fast and memory-efficient deployment while maintaining the training parallelism. Experiments show the competitive results.

**Strengths:**

* The target problem is of great interest to the Transformer community.

* The pioneering work to apply the retention mechanism to ViTs and make it work.

**Weaknesses:**

1. Although the motivation is to make a larger resolution possible. However, I do not find large-resolution experiments to support the claim. The main result table is still conducted with an image resolution of 224x224. Also, how about the real latency and memory costs? FLOPs may be misleading metrics as it only counts the computation numbers.

2. What are the difficulties when applying the retention mechanism to ViTs? Without sufficient discussion of the challenges and the corresponding opportunities, one can find it hard to identify the novelty of the proposed approach and think that it is a straightforward implementation of retention modules in ViTs.

3. Why does the image-based Transformer have the nature of autoregressive? It is understandable that NLP needs such autoregressive settings to predict future tokens with decoder-based models. While most ViTs are still encoder-based models without decoding stages. So I don't get the intention here why we need to do this.

**Questions:**

See weaknesses.